# Factors Associated with Decisions of Arab Minority Parents in Israel to Vaccinate Their Children against COVID-19

**DOI:** 10.3390/vaccines10060870

**Published:** 2022-05-29

**Authors:** Ola Ali-Saleh, Shiran Bord, Fuad Basis

**Affiliations:** 1Health Systems Management Department, The Max Stern Yezreel Valley College, Yezreel Valley 1930600, Israel; olaa@yvc.ac.il (O.A.-S.); shiranb@yvc.ac.il (S.B.); 2Faculty of Medicine, Technion Israel Institute of Technology, Haifa 2611001, Israel; 3Rambam Health Care Campus, Haifa 3109601, Israel

**Keywords:** Arab children, Israel, COVID-19 vaccine, parents’ intention, minority

## Abstract

The Arab ethnic community in Israel is characterized by low social economic status and is at risk due to the typically crowded households. Understanding parents’ level of awareness is important to avoid new outbreaks. Objectives: This study seeks to identify predicting factors associated with perceived susceptibility to COVID-19, and barriers to COVID-19 vaccination. Materials and Methods: A survey was conducted through social media, using snowball sampling via social networks. Additionally, *t*-tests, Chi-square tests, and *Z* tests were used to evaluate differences between independent proportions. Pearson correlations were calculated for the study variables. Multiple logistic regression examined the extent to which the background variables were related to the intention to vaccinate the child. Results: A total of 2843 Arab parents participated in the study. Older parents, higher socioeconomic status, higher trust in the authorities, vaccinated parents, and low psychological and physical barriers were positively correlated with willingness to vaccinate children. Pandemic fatigue was associated with less positive attitudes and reduced perceived effectiveness toward vaccination. Conclusion: Addressing minorities’ poor standards of living and physical and psychological barriers posed by the authorities to minorities’ access to vaccination may increase compliance with COVID-19 vaccination and protect the health of the entire population.

## 1. Introduction

The COVID-19 pandemic has global health and economic consequences [1]. From the first appearance of the virus in China in November 2019 until the end of 2021, 1,399,630 cases were diagnosed in Israel, with 8244 fatalities [2]. To prevent the spread of the virus and reduce its morbidity and mortality rates, countries worldwide implemented different strategies, including lockdowns, physical and social distancing, and the obligation to wear masks. 

The COVID-19 pandemic has had a deleterious impact on many aspects of children’s health, including bio-psycho-social damage such as anxiety disorders, eating disorders, behavioral changes, decline in learning abilities, and even a rise in domestic violence [3]. In a survey of social resilience conducted by the Israel Central Bureau of Statistics (2020), 25.8% of responding parents noted a deterioration in their children’s emotional state, along with an increase in consumption of food, sweets, and snacks and a decrease in physical activity, all of which may have a negative impact on children’s health [4]. Moreover, the increased use of virtual space among children necessitated by COVID-19 raises the risk of exposure to harm on the internet, including harassment, bullying, and social abuse [5].

The Ministry of Health (MOH) of Israel and the Israeli health management organizations (HMOs) launched a campaign to vaccinate the country’s populace on 19 December 2020 using the Pfizer-BioNTech COVID-19 vaccine. As of 9 May 2021, 5,421,711 Israelis (58.3%) had received the first dose of the vaccine and 5,078,537 (54.6%) had received both doses [6]. Research conducted in Israel examined the vaccine’s effectiveness two or three weeks after the first dose and a week after the second dose. After the first dose, the results showed a decline in the number of confirmed cases, the number of patients with symptoms, the number of hospitalizations, the number of patients with serious morbidity, and the number of deaths. Vaccine effectiveness rose after the second dose, showing a decline of 92% in confirmed cases, 94% in patients with symptoms, 87% in hospitalizations, 92% in serious morbidity, and 100% in mortality [7].

Concurrently, in January 2021, pediatricians in Israel reported a sharp rise in COVID-19 infections among children and adolescents, with 50,000 testing positive, a figure more than the total monthly figures in Israel during the first year and the first and second outbreaks [8]. While children are less severely affected by the disease, they are likely to be carriers, spreading the disease to others. Studies have shown that children are less contagious than adults, but the rate of transmission rises with age, such that the transmission rate among adolescents is high and similar to that among adults [9,10]. According to the Israeli Pediatric Association, as of April 2021, most confirmed cases of COVID-19 in Israel were children and adolescents. One out of every five confirmed case each day was between 10 and 18 years old, while the rates of COVID-19 infection among children aged 0–9 were significantly lower [11]. Therefore, the MOH intended to vaccinate children in schools. The logic behind the MOH’s decision was that in school, there is a great chance that children and youth sitting in the classrooms and walking around in breaks between lessons will infect each other, and these children constitute vectors or means of infection of their parents and grandparents. However, the Ministry of Education opposed this decision, and the decision was canceled [12]. 

Vaccinations were first approved for adolescents in August 2021 for children between the ages of 5 and 11. There were about 3.011 million children (age 0–17) in Israel in December 2021, who constituted 33% of the population. Of these, 2.173 million (72.2%) were Jewish and 735,000 (24.4%) were Arab [13]. The Arab population is generally younger than the Jewish population, with 43% under 18 (median age 22 years) compared to 28% of the Jewish population (median age 34 years) [14]. 

Morbidity and mortality rates have been observed to be especially high among ethnic minorities [15]. Moreover, due to their sociodemographic attributes such as crowded living conditions, high poverty rates, lack of supportive community infrastructures, and the closed nature of the society, ethnic minorities, such as the Arab population in Israel, are defined as communities at risk [16]. These attributes increase the risks to children during routine times, and even more so during the COVID-19 crisis. In 26.5% of Arab families there are more than two people per room, compared to 4.6% in the general Jewish population [17]. In 2016, 53% of Arab families lived below the poverty threshold, compared to 14% of Jewish families, and almost 66% of Arab children lived in poverty, compared to 20% of Jewish children [18]. 

In many groups in Arab society, large families with many children tend to live in close physical proximity. This proximity between older people and their children and grandchildren may lead to widespread viral infections [19]. 

Although there are many ways to prevent the spread of the virus, such as isolation, wearing masks, closing entertainment places, etc., it seems that vaccination is the most effective medical and public health strategy and the most successful policy in preventing morbidity and mortality from COVID-19, according to what is published in the literature [20]. Hence, parental compliance with children’s vaccinations is essential to achieving high vaccination coverage and reducing the prevalence of the infection.

The Israeli Pediatric Association and the Department of Infectious Pediatric Diseases (2020) supported vaccination of children aged 12–15 after the vaccine was approved, as initial results of stage III testing showed that the Pfizer-BioNTech COVID-19 vaccine exhibited 100% effectiveness against SARS-CoV-2 among children aged 12–15 [21].

The Health Belief Model (HBM) provides a basis for understanding decisions related to vaccination compliance in regard to perceived susceptibility (degree to which parents believe their children are at risk of contracting the disease), perceived severity (how dangerous is the disease for my children), perceived benefits of the vaccine, and perceived barriers to the vaccine (e.g., logistics barriers, fear of side effects, vaccine safety) [22]. According to this model, parents who believe their children are at high risk of a disease are more likely to vaccinate their children and recognize the benefits of vaccination [23]. Perceived barriers, such as fear of side effects and uncertainty regarding vaccination effectiveness, and perceived benefits of the vaccine are the strongest predictors of whether parents will vaccinate their children [24]. 

In addition, the Theory of Reasoned Action (TRA) claims that people’s intentions to carry out a particular health behavior are influenced by their attitudes toward subjective norms and behavior. Subjective norms are a consequence of people’s beliefs regarding how others want or expect them to behave, while people’s attitudes toward health behaviors are a result of their beliefs regarding the consequences of such behaviors [25]. Individuals will show positive attitudes toward a particular behavior if they believe that behavior will yield positive results [26]. Studies have found that compliance with children’s vaccinations is influenced by mothers’ personal opinions and past experience with vaccinations [24]. Moreover, a positive correlation was found between subjective attitudes and norms and compliance with childhood vaccinations [27]. 

Another element to be considered is “pandemic fatigue”. The World Health Organization (WHO) coined the term “pandemic fatigue” to express the widespread distress in response to an ongoing crisis whose end is not in sight, with people less motivated to implement recommended guidelines to protect themselves and others from the virus [28]. A cohort study conducted in Israel, which spanned three lockdown periods, showed that trust is a major component in public compliance. Fluctuations in risk perception and trust in government authorities were found to affect compliance with regulations. These instructions, among other things, might include the intention to get vaccinated. This study did not address the relation between pandemic fatigue and intention of citizens to get vaccinated, or of parents to vaccinate their children [29]. 

Vaccination of children is the next national task after vaccination of adults on the long road to achieving herd immunity. Vaccination programs are more successful when the rate of vaccination is high. To achieve this, it is important to understand the factors related to parental intentions to vaccinate children under the age of 18 against COVID-19.

## 2. Materials and Methods

### 2.1. Materials

This study is a cross-sectional survey. The survey questions were disseminated between 19 March and 2 April 2021 among Israeli Arab parents of children aged 6 months to 18 years using snowball sampling via social networks such as Instagram, Facebook, and WhatsApp. During the study period, vaccination for children aged between 16 and 18 began in Israel, while no recommendation was yet issued regarding vaccination of younger children. Participants were asked to pass the link to the survey questions to relatives and friends with children within the same age range. A total of 2483 parents responded and participated in the study. The study was approved by the Ethics Committee of Emek Yezreel College (approval no. YVC-EMEK 2021-52). 

### 2.2. Methods

The research tool is a quantitative, online, self-report questionnaire based on validated and reliable questionnaires used in previous studies which were adapted for the current study [29]. Questionnaire items in English were translated into Arabic and then back into English by an Arabic language expert to guarantee questionnaire reliability. The first page explained the research topic, assured the respondents of their anonymity, and informed them of their right to refuse to respond or withdraw from the study at any time without any consequences.

Questions related to perceived susceptibility, perceived severity, perceived benefits, and perceived barriers were taken from Prasetyo et al. [30], based on the Health Belief Model (HBM) [31].

Perceived susceptibility included three items, such as “The chance of my child getting infected by COVID-19 is high”. Perceived severity included three items, such as “COVID-19 is dangerous for my children and is liable to cause them suffering, serious complications and even death”. Perceived vaccine barriers included six items, such as “I am afraid the vaccine was developed too quickly, without information about safety or quality control in regard to children”. Perceived vaccine benefits included six items, such as “The COVID-19 vaccine will help return children to the regular school routine”.

Subjective attitudes and norms were tested based on items according to the Theory of Reasoned Action (TRA) [32], which was adapted to the current study. Subjective norms were evaluated by five items, such as “Most people I know plan to vaccinate their children”. Attitudes regarding vaccination were evaluated by five items, such as “I think COVID-19 is not dangerous for children so vaccination is unnecessary”. All the statements were rated on a five-point Likert scale, ranging from 1 (not at all) to 5 (to a large extent).

Nine statements were formulated to test pandemic fatigue. Eight were based on the WHO [28] recommendations for formulating policy to help countries deal with pandemic fatigue (e.g., “I’m tired of following instructions. The government/Health Ministry does not explain the logic behind the guidelines. Different government agencies convey conflicting messages about the guidelines”.). The ninth item referred specifically to fatigue from using Zoom (“I’m sick and tired of having the kids at home studying via Zoom”). Participants were asked to answer on a five-point Likert scale, ranging from 1 (do not agree at all) to 5 (very much agree).

In the demographic section of the questionnaire, participants were asked to provide the following personal details: age, sex, religion (Muslim, Druze, or Christian), extent of religious observance (secular, traditional, religious, very religious), marital status, number of children, residential area (north, south, center of Israel), and education (elementary—up to eight years; secondary—twelve years; post-secondary (vocational); undergraduate student; undergraduate degree; master’s degree or higher). 

Data analysis was performed using SPSS ver. 27. Internal consistencies were calculated for the study variables with Cronbach α, and the variables were composed with item means. Demographic and background variables were described with means and standard deviations, or frequencies and percentages, and compared to the intention to vaccinate the child with *t*-tests, Chi-square tests, and *Z* tests for the significance of the difference between independent proportions. Means, standard deviations, and Pearson correlations were calculated for the study variables. Multiple logistic regression was calculated to assess the extent to which the background variables and the study variables were related with the intention to vaccinate the child. The background variables were entered in step 1 and the study variables in step 2. Mediation was examined with the Process procedure [33], using model 4 for a dichotomous dependent variable and parallel mediation with four mediators. Continuous variables were standardized, and 5000 bootstrap samples were used with a 95% confidence interval. The Bonferroni criterion for multiple comparisons was applied per table.

The independent variables are those mentioned under the demographic variables above, such as: age, sex, religion, extent of religiousness, marital status, number of children, residential area, and education (see Table 1).The dependent variable was intention to vaccinate the child/ren (1) vs. hesitation/no intention to vaccinate the child (0). The Cronbach α of the attitude section of the questionnaire was α = 0.74; for subjective norms α = 0.79, for severity α = 0.72, for barriers α = 0.86, for effectiveness α = 0.91, and for pandemic fatigue α = 0.85. The r scoring for susceptibility = 0.57, *p* < 0.001. All variables were defined so that higher scores represented greater values of each phenomenon.

## 3. Results

Two thousand, four hundred and eighty-three (2483) Israeli Arab parents to minor children answered the questionnaire. Of these respondents, 91.7% were Muslim, 5.5% were Christians, and 2.8% were Druze. Additionally, 95.5% were mothers (4.5% fathers), 96.5% were married, and 96.1% had healthy children. Their age range was 20 to 58 years, with a mean age of about 33 years (*SD* = 7.15). Each parent had up to nine children (*M* = 2.65, *SD* = 1.38). Most parents (*n* = 1939, 78.1%) had more than one child, with the mean age of the oldest children being about nine years (*SD* = 5.60). For about a third of the parents (*n* = 851, 34.3%), the oldest child was over 12 years old and thus likely to be eligible for vaccination earlier than their younger siblings. 

About 46.4% of the parents (*n* = 1153) intended to vaccinate their children, while the others did not intend or were hesitant to do so (*n* = 1330, 53.6%). About 77% of the parents were vaccinated against the COVID-19 virus, 12% had been infected by the virus, and 11% chose not to be vaccinated (Table 1).

About 51% of the parents had academic education, 9% had at least an undergraduate degree, 18% had high school education, and about 21% had up to 12 years of education. About 55% of the participants lived in rural areas, mostly in the northern part of Israel (61%). About 57% of them were religious, and most of the rest (37%) were partly religious. The economic status in 47% of the cases was low, in 39% it was average, and in 14% it was good (see Figure 1 for details).

Applying the Bonferroni criterion for multiple comparisons, some significant differences were found in the intention to vaccinate the children among different groups (Table 1, *p* < 0.004). Age, number of children, age of the oldest child, higher economic status, and parent vaccination were related to the intention to vaccinate the children. In that regard, parent age, number of children, and age of the oldest child were highly inter-correlated (*r* = 0.58 to *r* = 0.78, *p* < 0.001). Parent age and number of children were highly correlated with the oldest child being over 12 years old (*r* = 0.55 and *r* = 0.71, *p* < 0.001). To conclude, in light of the significant differences presented in Table 1, control variables used in further analyses were parent age, economic status (1—very bad, 5—very good), and parent vaccination status (1—yes, 0—no).

Distributions in Table 2 reveal that most of the study variables—namely attitudes toward vaccination, subjective norms, susceptibility to the virus, perceived severity of the disease, barriers to vaccination, and pandemic fatigue—had moderate means. A moderate–high average score was found for effectiveness of the vaccination. 

The Bonferroni criterion for multiple comparisons revealed a significant correlation between the study variables (*p* < 0.001). Intention to vaccinate the children was positively and highly related to attitudes toward vaccination, subjective norms, and effectiveness of the vaccination. Weaker yet positive significant correlations were found between intention to vaccinate the children and perception of susceptibility and severity of the disease. There was a negative correlation between willingness to vaccinate the children and pandemic fatigue and physical barriers to vaccination. 

Multiple logistic regression was calculated to assess the extent to which the background variables and the study variables (attitudes toward vaccination, subjective norms, susceptibility to the virus, perceived severity of the disease, barriers to vaccination, effectiveness of the vaccination, and pandemic fatigue) were related with intention to vaccinate the children against COVID-19 (Table 3). The resulting model was found significant, which explains about 64% of the variance in intention to vaccinate the children. Applying the Bonferroni criterion for multiple comparisons (*p* < 0.005) revealed that of the background characteristics, age and parent vaccination were significant, as older parents and parents who were vaccinated showed higher odds for positive intention concerning vaccination of their children. In addition, higher positive attitudes toward vaccination, higher supportive subjective norms, higher perceived susceptibility to the virus, lower barriers to vaccination, and greater perceived effectiveness of the vaccination were related to higher odds for intention to vaccinate the children.

To assess the extent to which attitudes toward vaccination, subjective norms, barriers to vaccination, and perceived effectiveness of the vaccination mediated the relationship between pandemic fatigue and intention to vaccinate the children, a Process model [33] was applied, using model 4 for a dichotomous dependent variable, with parallel mediation. Continuous variables were standardized, and the control variables were age, economic status, and parent vaccination (Figure 2). The total indirect effect was found significant (effect = −0.89, *SE* = 0.06, 95% *CI*: −1.02, −0.77), as well as all four specific indirect effects (trust: attitudes = −0.31, *SE* = 0.03, 95% *CI*: −0.37, −0.25; subjective norms = −0.16, *SE* = 0.02, 95% *CI*: −0.21, −0.12; barriers: effect = −0.31, *SE* = 0.03, 95% *CI*: −0.38, −0.24; and effectiveness of the vaccination = −0.12, *SE* = 0.02, 95% *CI*: −0.16, −0.08). This indicates that higher pandemic fatigue was related with less positive attitudes toward vaccination, less supportive subjective norms, higher barriers to vaccination, and lower perceived effectiveness of the vaccination, which were then related with lower odds for intention to vaccinate the children.

## 4. Discussion

The COVID-19 pandemic has had severe health and economic impacts worldwide [1]. The morbidity and mortality rates have been especially high in ethnic minority groups [15]. Our study focused on the Arab minority in Israel. In our study, 91.7% of the respondents were Muslims, as they are the majority of Arabs in Israel (96%), 5.5% were Christians, and 2.8% were Druze. Most participants had more than a high school education (78.8%), and 95.5% of the respondents were women (mothers). Our questionnaire was distributed through Social Networking Services (SNS). Some studies have shown that women tend to use SNS more than men [34,35]. In their study, Araz et al. showed that in the early stages of the COVID-19 epidemic, the level of anxiety among people who used SNS was significantly higher than that of those who did not [36]. 

To control for the fact that most parents responded generally in relation to having several children and for the fact that in about a third of the families children would be eligible for vaccination earlier than in other families, parent age was controlled for in all further multivariate analyses. Economic status was controlled for as well. Economic status did not deviate from a normal distribution (five categories, skewness = 0.20, SE = 0.05) and was thus used as a continuous variable.

According to the current study results, the older the children, the higher the intention to vaccinate them. This may be explained partially by fear of parents to vaccinate their younger children, as the safety of the vaccines in the official media in the beginning, and in SNS later on, was controversial. Furthermore, parents might have considered their younger children’s fear of needle stick, according to their previous experience with drawing blood for tests or with influenza vaccination. Some studies have even suggested ways parents should prepare themselves before presenting their children for COVID-19 vaccination [37].

Although the socioeconomic status of Arabs in Israel has improved in recent years, there are still significant gaps between the conditions of the Arabs and the Jews. In about 26.5% of Arab families, more than two people live in one room, compared to 4.6% in the general Jewish population [17]. In 2016, 53% of Arab families and almost 66% of Arab children lived below the poverty line, compared to 14% of Jewish families and 20% of Jewish children [18]. The gaps between the Arab and Jewish populations in Israel may stem from pre-labor-market barriers to investment in human capital, such as low level of transportation/infrastructure, shortage of accessible employment areas, barriers to market entry, and weakness of Arab local authorities [38]. Low socioeconomic status is well correlated with low compliance to vaccination, which coincides with the findings of Daoud et al. about the influence of social, educational, and material conditions of the Arab minority in Israel on their low health awareness [39]. Vaccination is the most effective medical and public health strategy and the most successful policy in preventing morbidity and mortality of diseases whose vaccines are available. Hence, parental compliance with children’s vaccination is essential to achieving high vaccination coverage and reducing the prevalence of diseases [20]. The present study demonstrates that vaccine hesitancy led to half of the Arab parents in the study not administering the anti-COVID-19 immunization to their children. In a previous study, we found that despite the success of the national vaccination campaign, the Arab population was still at the bottom of the list among Israeli citizens, and in some predominantly Arab localities (Bedouin) the response to vaccination was nil. Reports from the MOH on 1 February 2021 showed that approximately 32% of the total population of Israel had received their first vaccination dose, compared to 18% of the Arab section [40]. In addition, 81% of the total population in Israel over the age of 60 had received the first dose of the vaccine, compared to 57% among the Arab population [41]. Social norms influenced by living in the same village, closed communities at the periphery, and psychological and physical barriers might have influenced the overall willingness of Arab parents to vaccinate their children.

There were some challenges facing parents who did not intend to vaccinate their children. The major ones are related to how they intend to prevent the spread of COVID-19 among their parents and grandparents who live in close proximity to their houses and neighborhoods. In addition, some of them avoided children’s vaccination to save pain and struggle with their younger children. Furthermore, those parents who did not choose to vaccinate their children were asked to provide a negative fast antigen kit every third day to their teacher at school, otherwise, they were not allowed to inter the school. On the other hand, those who chose to vaccinate their children had a struggle with who did not intend to vaccinate their children. There was great argument in social networks among these two groups. These parents were afraid their children would get infected and infect them, especially when new more contagious mutations emerged. 

Our study showed significant correlations between age of the parents and the intention to vaccinate their children. In a previous study, we found higher vaccination rates among elderly Arabs, as they perceived the disease as more dangerous to them [41]. Moreover, attitudes toward vaccination, supportive subjective norms, perceived susceptibility to the virus, and perceived effectiveness of the vaccine (benefits) were positively related to intention to vaccinate the children. Intention to vaccinate the children might be affected by inter-generational households in a society where there are interfamily gatherings and close family relations within extended Arab families living in the same village, as it is pertinent to protect the elderly by vaccinating the little ones.

Although Ruggiero et al. recently in their study mentioned that lower education was correlated with lower intention to vaccinate children with routine vaccination [42], and in our previous study, we reported that a higher level of education was well correlated with higher intention to get vaccinated [41], in this study we did not find a significant difference in the intention to vaccinate children between parents with low education and parents with high education. Perhaps other factors play rule in the intention to vaccinate children against COVID-19, such as emotional and social norms, misinformation about the safety of vaccine among children, etc. 

A positive correlation was found between subjective attitudes/norms and compliance with child vaccination. Similar findings were made by Pot et al. in their study of the impacts of parents’ attitudes, beliefs, subjective norms, and perceived vaccination effectiveness on the intentions to vaccinate children against the human papilloma virus (HPV) [43]. 

Psychological and physical barriers to vaccination were negatively related to the intention to vaccinate the children. Following the problems of accessibility of vaccination centers by the Arab elderly population (physical barriers), the MOH overcame this barrier, but the psychological barriers remained. In previous studies related to COVID-19 vaccination for the elderly, it has been shown that mistrust of the benefits and safety of the vaccines and concerns about their unintended effects are key barriers to willingness to be vaccinated [44,45]. Other studies have mentioned the mistrust of COVID-19 vaccines among people of color. It is believed that mistrust is multifactorial and not restricted to concerns about COVID-19 vaccine safety and efficacy. It is rooted in a history of unethical medical and public health experience involving communities of color, as well as structural inequities in government institutions (for example, the police, criminal justice, child welfare, and public schools). As a result, a primary strategy to decrease mistrust has been to get trusted community leaders to engage communities of color in public health campaigns [42,46]. The same conclusion may apply to the Arab community in Israel. 

COVID-19 has been around for more than two years, with decreasing perceived risk of the disease as people become accustomed to it [28]. Understanding how the pandemic and the restrictions imposed have affected people’s everyday lives, their social and mental health, and their motivation and intentions to follow recommended practices is critical for the sustained success of stopping the progression of the pandemic. When difficult social and economic circumstances and unclear guidelines persist, it is natural to expect a decline in people’s trust in authorities and motivation to comply with guidelines to prevent disease spread, resulting in pandemic fatigue and low motivation [47]. 

Mahmood et al. conducted a comprehensive study to investigate the impact of the COVID-19 pandemic on ethnic minorities in the UK with mainly African and Asian roots. Among other things, they investigated factors that caused emotional fatigue in these ethnic minority communities. In addition, living close to each other made it difficult to wear masks and avoid physical contact. Low socioeconomic status, social norms, social boundaries, fear of stigma of getting infected, and being socially and physically isolated led to considerable mental stress, emotional fatigue, and less adherence to procedures. Since most Arabs in Israel live in similar conditions, it may be concluded that they are more prone to pandemic fatigue than urban residents, who live in better socioeconomic conditions, with less demanding relations with neighbors and less strict community traditions and beliefs [48]. This issue is reflected in our model (Figure 2).

Although the findings of the present study show that pandemic fatigue is not directly related to parents’ intention to vaccinate their children, it appears to be indirectly related to this decision through a variety of variables directly related to intention to vaccinate children as described above (negative correlation with perception of vaccine effectiveness, social norms, and attitude to the vaccine; positive correlation with psychological and physical barriers). These findings are of great importance, as they teach about the many factors to which pandemic fatigue is linked and emphasize the need to address them. Dealing with pandemic fatigue requires a robust and multipronged response that addresses motivation in terms of costs and benefits of mitigation behaviors. Many are unable to work from home, they are taking care of isolated children, businesses are slow or closed, and people are separated from their loved ones. Thus, the value in complying with safety guidelines for a long time, discounted by mental fatigue and even depression, is increasingly outweighed by the value in not doing so [49]. This issue is worth investigating further. Long periods of restrictions, with all their consequences, should be weighed against pandemic fatigue and low compliance with instructions and vaccinations. One may wonder how the entire world lived through the H_1_N_1_ pandemic in 2008–2009 without strict restrictions, although the disease was more dangerous to younger sections of the population [50]. The authors believe this dilemma will be dealt with after COVID-19 becomes endemic rather than a pandemic. 

There are a few limitations of this study. The questionnaire used focused on the items relating to intention to vaccinate against COVID-19, but it did not take into consideration the correlation between our results and intention to vaccinate children during previous flu seasons. Furthermore, the fact that the survey took place at a time Israel had succeeded in decreasing the spread of the virus, thanks to an intense adult vaccination program, might have influenced the results. In addition, when performing the study, the hesitancy toward vaccination of children against COVID-19 was still a matter of debate in the official media. The results might have been different if there was no debate whatever about the safety of the vaccines for children. There may be need for further studies when the vaccines have been proved to be certainly safe for children. 

## 5. Conclusions

Culture, socioeconomic status, education, and psychological and physical boundaries may influence the attitude, norms, and willingness of minority groups to vaccinate their children against COVID-19. Governments should increase minorities’ involvement and trust in government and professional authorities. This should be achieved not only in pandemics, but rather in every day. It is well known that the political background in Israel had created mistrust between both parties Authorities should avoid long fluctuations and unstable clear instructions with contradictory restrictions that may lead to pandemic fatigue and mistrust. Moreover, authorities should internalize that the wellbeing of the entire population, particularly during pandemics, depends on the cooperation and trust of all minorities in the country, as “The chain strength is strong as its weakest link”. It is hoped that politicians and decision-makers will consider the points raised in this paper for effective management of pandemics.

## Figures and Tables

**Figure 1 vaccines-10-00870-f001:**
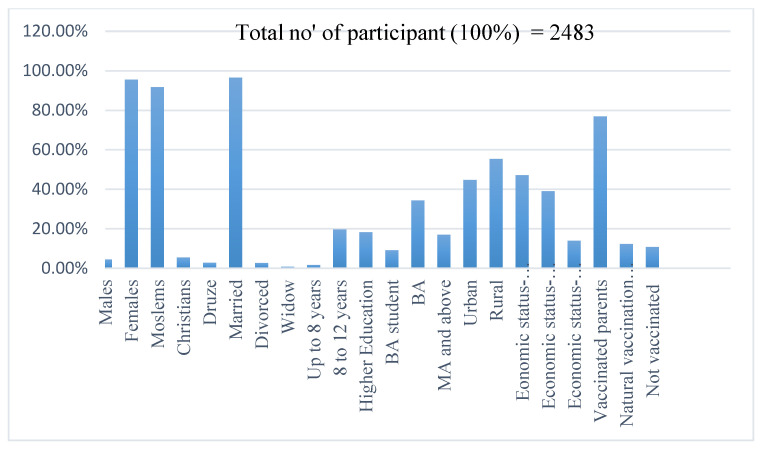
Demographic data of participants. Percent of each sector from total number of participants.

**Figure 2 vaccines-10-00870-f002:**
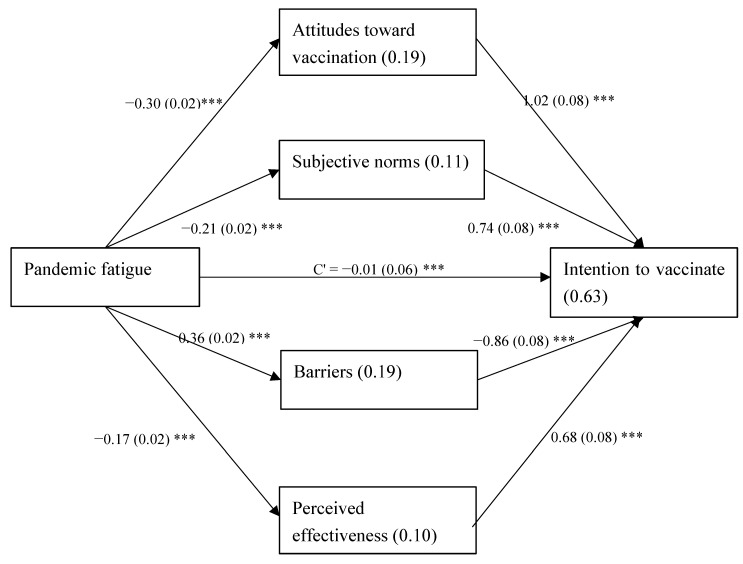
Process model for the indirect effect between pandemic fatigue and the intention to vaccinate the child/ren. Note: Values on arrows: *B*(*SE*), values within rectangles: *R*^2^, C’ = direct effect. *** *p* < 0.001.

**Table 1 vaccines-10-00870-t001:** Background characteristics by the intention to vaccinate (*N* = 2483).

Sociodemographic and Background Characteristics				
	Total Sample	No Intention/Hesitation (*n* = 1330)	Intention to Vaccinate (*n* = 1153)	
Gender (%)				
Male	112 (4.5)	56 (50.0)	56 (50.0)	*Z* = 0.77 *p* = 0.439
Female	2371 (95.5)	1274 (53.7)	1097 (46.3)	
Mean age (*SD*), range	33.07 (7.15), 20–58	31.59 (6.43)	34.77 (7.55)	*t*(2276) ^(1)^ = 11.21 *p* < 0.001
Religion (%)				
Muslim	2277 (91.7)	1221 (53.6)	1056 (46.4)	χ^2^(2) = 0.06 *p* = 0.970
Christian	137 (5.5)	73 (53.3)	64 (46.7)	
Druze	69 (2.8)	33 (52.2)	33 (47.8)	
Marital status (%)				
Married	2397 (96.5)	1292 (53.9)	1105 (46.1)	χ^2^(2) = 3.68 *p* = 0.159
Divorced	67 (2.7)	31 (46.3)	36 (53.7)	
Widow	19 (0.8)	7 (36.8)	12 (63.2)	
Mean number of children (*SD*), range	2.65 (1.38), 1–9	2.50 (1.36)	2.82 (1.37)	*t*(2481) = 5.94 *p* < 0.001
Mean age of oldest minor child (*SD*), range	8.73 (5.60) 0.5–18	7.65 (5.27)	9.97 (5.73)	*t*(2359.75) ^(1)^ = 10.50 *p* < 0.001
Education (%)				
Up to 8 years	42 (1.7)	22 (52.4)	20 (47.6)	*Z* = 0.36 *p* = 0.718 (academic vs. non-academic)
8 to 12 years	486 (19.6)	265 (54.5)	221 (45.5)	
Higher Education	451 (18.2)	233 (51.7)	218 (48.3)	
BA student	229 (9.2)	138 (60.3)	91 (39.7)	
BA	854 (34.4)	474 (55.5)	380 (44.5)	
MA and above	421 (17.0)	198 (47.0)	223 (53.0)	
Area of residence (%)				
Northern Israel	1509 (60.8)	788 (52.2)	721 (47.8)	χ^2^(2) = 3.8 *p* = 0.148
Central Israel	805 (32.4)	442 (54.9)	363 (45.1)	
South Israel	169 (6.8)	100 (59.2)	69 (40.8)	
Type of residence (%)				
Urban	1110 (44.7)	598 (53.9)	512 (46.1)	*Z* = 0.28 *p* = 0.781
Rural	1373 (55.3)	732 (53.3)	641 (46.7)	
Level of religiosity (%)				
Secular	145 (5.8)	69 (47.6)	76 (52.4)	χ^2^(3) = 3.75 *p* = 0.289
Partly religious	925 (37.3)	485 (52.4)	440 (47.6)	
Religious	1318 (53.1)	722 (54.8)	596 (45.2)	
Orthodox	95 (3.8)	54 (56.8)	41 (43.2)	
Economic status (%)				
Below average	1170 (47.1)	690 (59.0)	480 (41.0)	χ^2^(2) = 26.97 *p* < 0.001
About average	969 (39.0)	480 (49.5)	489 (50.5)	
Above average	344 (13.9)	160 (46.5)	184 (53.5)	
Parent vaccination (%)				
Yes, or has an appointment	1910 (76.9)	879 (46.0)	1031 (54.0)	χ^2^(2) = 205.38 *p* < 0.001
Was sick with COVID-19	306 (12.3)	217 (70.9)	89 (29.1)	
No	267 (10.8)	234 (87.6)	33 (12.4)	
Child’s health (%)				
Healthy	2386 (96.1)	1284 (53.8)	1110 (46.2)	*Z* = 1.24 *p* = 0.216
Not healthy	97 (3.9)	46 (47.4)	51 (52.6)	

^(1)^—t for unequal variances.

**Table 2 vaccines-10-00870-t002:** Means, standard deviations, and correlations between the intention to vaccinate children and other study variables (*N* = 2483).

	*M* (*SD*)	2.	3.	4.	5.	6.	7.	8.
1. Intention to vaccinate	0.46 (0.50)	0.59 *	0.51 *	0.16 *	0.23 *	−0.52 *	0.49 *	−0.21 *
2. Attitudes	3.40 (1.04)		0.50 *	0.15 *	0.40 *	−0.57 *	0.50 *	−0.34 *
3. Subjective norms	3.01 (0.90)			0.18 *	0.27 *	−0.45 *	0.54 *	−0.24 *
4. Susceptibility	3.24 (1.15)				0.28 *	−0.04	0.29 *	−0.01
5. Severity	3.17 (1.00)					−0.12 *	0.32 *	−0.20 *
6. Barriers	3.37 (0.99)						−0.37 *	0.39 *
7. Effectiveness	3.76 (0.97)							−0.21 *
8. Pandemic fatigue	3.37 (0.92)							

* *p* < 0.001. Note—range (except for intention): 1–5.

**Table 3 vaccines-10-00870-t003:** Multiple logistic regression for the intention to vaccinate the child/ren (*N* = 2483).

	*B* (*SE*)	*OR* (95% *CI*)	*p*
**Step 1**			
Age	0.06 (0.01)	1.06 (1.05, 1.08)	<0.001
Economic status (higher)	0.09 (0.04)	1.09 (1.01, 1.19)	0.029
Parent vaccination (yes)	1.43 (0.11)	4.19 (3.35, 5.24)	<0.001
**Step 2**			
Age	0.06 (0.01)	1.06 (1.04, 1.08)	<0.001
Economic status (higher)	0.03 (0.05)	1.03 (0.93, 1.13)	0.595
Parent vaccination (yes)	0.95 (0.16)	2.59 (1.91, 3.53)	<0.001
Attitudes	1.00 (0.08)	2.73 (2.32, 3.21)	<0.001
Subjective norms	0.83 (0.09)	2.29 (1.92, 2.74)	<0.001
Susceptibility	0.17 (0.06)	1.18 (1.06, 1.33)	0.003
Severity	0.16 (0.06)	1.17 (1.03, 1.33)	0.016
Barriers	−0.88 (0.08)	0.42 (0.35, 0.49)	<0.001
Effectiveness	0.67 (0.08)	1.95 (1.66, 2.28)	<0.001
Pandemic fatigue	−0.03 (0.07)	0.97 (0.85, 1.11)	0.632

Step 1: χ^2^(3) = 313.88, *p* < 0.001, Nagelkerke’s *R*^2^ = 0.159; Step 2: χ^2^(7) = 1294.76, *p* < 0.001, Nagelkerke’s Δ*R*^2^ = 0.478; Total model: χ^2^(10) = 1608.64, *p* < 0.001, Nagelkerke’s *R*^2^ = 0.637.

## Data Availability

All data generated or analyzed during this study are included in this published article.

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
