# Peer review of "Factors Associated with Decisions of Arab Minority Parents in Israel to Vaccinate Their Children against COVID-19"

_vaccines, 2022, doi:10.3390/vaccines10060870_

Round 1

Reviewer 1 Report

The manuscript entitled “factors associated with decision/intentions or Arab minority parents in Israel to vaccinate their children against COVID-19” y Ali-Saleh in timely important investigation and innovative aspect of COVID-19 and vaccination scenario. The author conducted a detailed study, and the results are discussed accordingly. This manuscript is suitable for publication in an esteemed vaccines journal. However, before to accept for publication, the author needs to address the following comments

  1. The affiliation of the study is not clear. The author must provide full details and official email addresses
  2. The author should reduce the abstract size and keep the most valuable information
  3. I appreciate the authors for the excellent introduction
  4. It will be great to divide the section 2, materials, and method into subsections accordingly
  5. There are several tables, the author should attempt to make some information from the table present in the form of a figure/scheme
  6. The author should incorporate a scheme to briefly present the overview of the study
  7. The author should look at the similar types of studies conducted on various aspects and discuss them
  8. What are the major challenges for Arab minority parents in Israel not encouraging vaccination and if encouraging what is the reason?
  9. Is any impact of education on vaccination
  10. What is the major solution to increase vaccination?
  11. Lack of vaccination to children effecting on their schooling or not

Author Response

Dear reviewer #1

Thank you your valuable comments. We hope we addressed all your comment in the text with Track changes. We also wrote notes mentioning your note number (in black font color). We also answered to each comment below and referred you to specific line. We hope our changes and additions meat your expectations.

Thank you again

Fuad Basis – the corresponding author

Reviewer 2 Report

Thank you very much for the opportunity to review this paper which addresses the COVID-19 vaccination issues from an interesting perspective in terms of the population studied. I appreciated particularly the clarity with which the authors reported the Methods and Results. Discussion is appropriate as is the number of references cited. My only concern is about Introduction, which needs to be shortened and made more focused. Minor English spell checks are required.

Author Response

Reviewer #2 comments

Comments and Suggestions for Authors

Dear reviewer #2

Thank you your valuable comments. We hope we addressed all your comment in the text with Track changes. We also wrote notes mentioning your note number (in red font color). We also answered to each comment below and referred you to specific line. We hope our changes and additions meat your expectations.

Thank you again

Fuad Basis – the corresponding author

Reviewer 3 Report

This is a  very well-designed study focusing on various factors that are linked with taking decisions of the Israeli Arab minority to take covid-19 vaccine for children. The study included a standard questionnaire to get the relevant data for analysis. All the data were analyzed using robust appropriate statistical methods. The result, discussion, and conclusion are well stated reflecting the findings of the study.  However, I have a few comments as follows:

Comments:

The title has to be specific; use either decision or intentions, I suggest to use decision. Not both!

Please join 1st and 2nd para of the introduction to make a single paragraph in the introduction

Line 95. What are the other strategies to control covid than the vaccine, please write a few of them here like using mask, avoiding crowded places etc…

Provide the population size questioned/surveyed in this study under methodology. e.g., 2483

Line 200, what were independent variables,  mention it here.

Line 208. 98.7% were Muslim, so what were the remaining %?? Christian. Jewish??

In many cases, discussions are being written under the result section. The result should only focus on what was found, the actual result, not anything else, for example line  233 to 240, line 288-269 and so …move these under discussion where applicable.

Thanks for focusing on the limitations of the study.

Author Response

Answers to Reviewer #3 comments

Dear reviewer #3

Thank you your valuable comments. We hope we addressed all your comment in the text with Track changes. We also wrote notes mentioning your note number (in blue color). We also answered to each comment below and referred you to specific line. We hope our changes and additions meat your expectations.

Thank you again

Fuad Basis – the corresponding author
